# DISCRETE PREDICTIVE REPRESENTATION FOR LONG-HORIZON PLANNING

## ABSTRACT

Discrete representations have been key in enabling robots to plan at more abstract levels and solve temporally-extended tasks more efficiently for decades. However, they typically require expert specifications. On the other hand, deep reinforcement learning aims to learn to solve tasks end-to-end, but struggles with long-horizon tasks. In this work, we propose Discrete Object-factorized Representation Planning (DORP), which learns temporally-abstracted discrete representations from exploratory video data in an unsupervised fashion via a mutual information maximization objective. DORP plans a sequence of abstract states for a low-level model-predictive controller to follow. In our experiments, we show that DORP robustly solves unseen long-horizon tasks. Interestingly, it discovers independent representations per object and binary properties such as a key-and-door.

## 1 INTRODUCTION

In future, we hope that robots will be able to operate in unstructured environments such as homes and hospitals, and endowed with long-horizon planning ability. Despite successes in deep reinforcement learning (RL) from raw observations, much progress relies on the availability of shaped reward to guide the learning (Ng et al., 1999; Mirza et al., 2020). On the other hand, over past decades, task and motion planning has been shown to solve much longer-horizon goal-directed tasks such as making a cup of coffee from torque control (Kaelbling & Lozano-Pérez, 2011; Srivastava et al., 2014; Toussaint, 2015; Wang et al., 2018). However, these methods often require pre-specified discrete abstract states, task representations and transition models, e.g., whether the robot is holding a cup and what actions (or perturbations) change such an abstract state. In this paper, we aim to learn discrete representations for high-level abstract planning from video interaction data, combined with a learned short-horizon controller.

Learning discrete representations from unsupervised data for planning is challenging for two reasons. First, the relationship between the optimization objective and the true task objective is not well-defined. Second, optimizing a model with a discrete layer is difficult with standard deep learning techniques. Recent methods approach the first problem using reconstruction or constrastive objectives (Watter et al., 2015; Anand et al., 2019; Ha & Schmidhuber, 2018; Kurutach et al., 2018; Hafner et al., 2019; Srinivas et al., 2020); however, the learned representations are in continuous latent space which is unstructured and difficult to combine with high-level abstract planning. While other methods show promise in learning discrete representations, they have not been applied to temporally-extended RL tasks (Oord et al., 2018; Razavi et al., 2019; Risi & Stanley, 2019; Asai & Fukunaga, 2017; Stratos & Wiseman, 2020).

In this work, we propose Discrete Object-factorized Representations for Planning (DORP) – a novel framework for visual planning and control by learning discrete representations and a low-level controller. DORP learns discrete representations from images that change slowly overtime, such as whether or not the agent holds a key or which room the agent is in, along with a low-level predictive model for control. These slow features enable the agent to plan at a low frequency in longer-horizon tasks. More specifically, DORP represents an abstract state as a set of one-hot vectors, and optimizes its encoder by maximizing a mutual information lower bound between the current representations to future observations (Oord et al., 2018). In order to train through the discrete layer, we apply the Gumbel-Softmax reparametrization trick (Jang et al., 2016; Maddison et al., 2016). Using abstract states as nodes, we build an approximate feasibility graph based on observed transition data. When provided with new start and goal images, the agent plan the shortest abstract path. Using the next

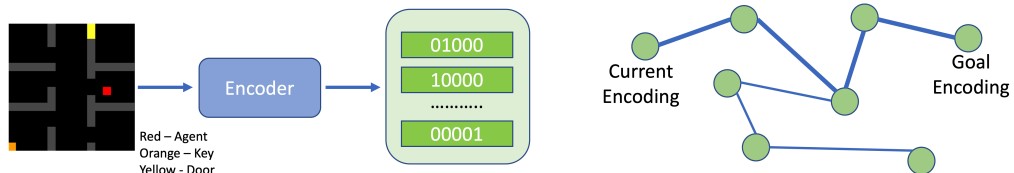

(a) Unsupervisedly encode image with one-hot encodings        (b) Plan on the discrete encoding space with graph search

Figure 1: Learning discrete representations for planning. We approach the long-horizon planning on high dimensional image space by learning a discrete representation and planning on the discrete space. (a) We learn multiple one-hot encodings for each observation fully unsupervisedly with contrastive learning. Each one-hot encoding corresponds to a temporally abstract state of one of the freely moving entity in the observation, such as a room the agent is in or whether a key has been picked up. (b) With the discrete encoding, we build a graph that connects the current encoding to the goal encoding and do a graph search on it to plan efficiently for long horizon tasks.

abstraction as waypoint, model-predictive control maximizes the objective that is 1 if it reaches the target abstraction and 0 otherwise with a trained video prediction model. Unlike other subgoal planning work (Savinov et al., 2018; Nasiriany et al., 2019; Laskin et al., 2020; Liu et al., 2020), by following abstract waypoints, DORP avoids unneccessary steps to match exact waypoint states.

In a set of experiments, we demonstrate that DORP learns temporally-consistent and object-factorized representations suitable for planning. We show that these representations enable DORP to handle unseen long-horizon tasks more successfully compared to the states-of-the-arts in visual planning. Interestingly, we observe that latent representations show object-level factorization such as key-and-door.

## 2 PRELIMINARIES

We present background material on unsupervised representation learning and discrete optimization that our method builds on, and our problem formulation.

**Contrastive Predictive Coding (CPC)** (Oord et al., 2018) learns low-dimensional representations that are most predictive of the future future high-dimensional sequential data. A non-linear encoder $q_\theta : \mathcal{O} \to \mathbb{R}^l$, parametrized by $\theta$, encodes the observation $o_t \in \mathcal{O}$ to a latent $l$-dimensional vector representation $z_t$. Let's define a similarity score $f_k(z_t, o_{t+k}) = \exp(z_t \psi q_\theta(o_{t+k}))$ where $\psi$ is a trainable $l$-by-$l$ similarity matrix and $o_{t+k}$ is a future observation $k$ steps ahead of $o_t$. Given the *query* observation $o_t$, we aim to classify the *key* observations – $o_{t+k}$ as positive and other sample $\tilde{o}$ from the dataset as negative. Formally, we optimize the loss function $\mathcal{L}_{CPC} = -\mathbb{E}_{o_t, o_{t+k}} \left[ \log f_k(z_t, o_{t+k}) - \log \sum_{o_j \in X} f_k(z_t, o_j) \right]$ with respect to $\theta$ and $\psi$. This also corresponds to maximizing a lowerbound of the mutual information between the latent representation $z_t$ and the future observation $o_{t+k}$.

**Gumbel-Softmax (GS)** (Jang et al., 2016) is a discrete optimization technique to compute the gradient of through samples from a categorical distribution $\pi_i$. GS first applies a reparametrization trick to rewrite samples $z_i$ as onehot($\arg\max_j(g_j + log\pi_i)$) where $g_j$ are i.i.d samples from Gumbel(0, 1) (Gumbel, 1935). GS approximates $\arg\max$ with softmax making the discrete stochastic layer differentiable, i.e., $z_i = \text{softmax}((g_i + \log \pi_i)/\tau)$. When the temperature parameter $\tau$ approaches 0, the samples converge to the true categorical distribution. Empirically, the temperature $\tau$ starts high and is annealed closer to 0 by a certain schedule.

### 2.1 PROBLEM STATEMENT: GOAL-DIRECTED VISUAL PLANNING AND CONTROL

We define an unknown, fully-observable, stochastic dynamical system $f$ which maps input observation $o_t \in \mathcal{O}$ and action $a_t \in \mathcal{A}$ to the next observation $o_{t+1}$. Under this dynamical system, we assume a simple exploration policy $\pi_{rand}$ which can collect data that characterizes the dynamics of the system. This is known as self-supervised data or play data (Agrawal et al., 2016; Lynch et al., 2020). We consider high-dimensional observation such as images.

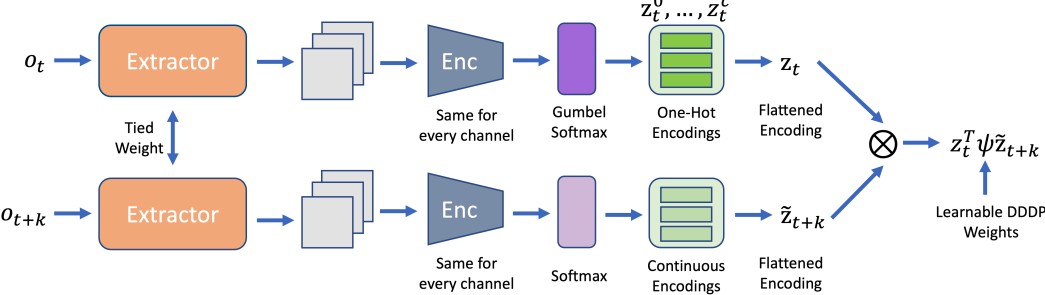

Figure 2: Unsupervised discrete code learning. We propose to learn a set of one-hot encodings as the latent representation for each observation. For an anchor observation $o_t$, its neighbours $o_{t+k}$ for some small $k$ are treated as positives, and random other observations are negatives. We propose a learnable weight matrix to be diagonally dominant and diagonally positive (DDDP). An object extractor architecture is applied to learn to extract objects and a shared encoder is applied per channel. Finally, Gumbel softmax and softmax are used to maximally allow gradient to flow back to the encoder.

Our goal is to learn discrete abstraction $z_t \in \mathcal{Z}$ given observation $o_t$, an abstract feasibility model, and a low-level local controller. At test time, given start $o_s$ and goal $o_g$ observations, we can plan a sequence of abstract states $z_s, z_1, ..., z_g$ where $z_s$ and $z_g$ are the representation of $o_s$ and $o_g$, and apply the low-level controller to reach these abstract waypoints and finally to $o_g$.

## 3 DISCRETE OBJECT-FACTORIZED REPRESENTATION FOR PLANNING

Our objective is to derive representation properties such as *object-factorization* and *temporal-consistency* from an unsupervised learning objective to facilitate long-horizon planning. When object representations are factorized, the agent can escape the need of combinatorial data configurations and can quickly generalize to unseen tasks. Also, the connectivity graph can also be memory-efficiently represented and combined with a more powerful planning algorithm. Another important property for a representation is temporal consistency, meaning any two state observations in the same abstraction should be reachable from one-another by a short sequence of actions. When this property holds in the latent space, a high-level plan can be successfully executed by a low-level controller.

We propose Discrete Object-factorized Representation for Planning (DORP) to tackle goal-directed visual planning and control problem. DORP learns latent discrete representations, a connectivity graph for abstract planning, and a low-level model-predictive controller. We describe each subcomponent in Section 3.1 - 3.3.

### 3.1 DISCRETE REPRESENTATIONS

Despite successes in applying its unsupervisedly learned representations for downstream tasks, CPC is not endowed with an inductive bias for its representations to facilitate planning (Oord et al., 2018). Additionally, it is unclear how a discrete representations can be extracted and optimized. Toward this goal, we propose an object-factorized architecture with asymmetric Gumbel softmax for discrete optimization, and a diagonally dominating and diagonally positive similarity matrix to encourage temporal consistency.

We implement our DORP architecture to encourage object-factorization and capture discrete representations. The extractor passes an anchor observation through a convolutional neural network and outputs a $c$-channel feature map. Each feature map is inputted to a shared encoder followed by a Gumbel softmax. For a positive and negative pair of images, however, we opt for a continuous embedding by swapping Gumbel softmax for softmax. Empirically, without this asymmetry trick, the optimization tends to converge to a poor local optima. Together these one-hots are flattened into long vectors, and their bilinear product is the similarity score between the query-key pair (see Figure 2).

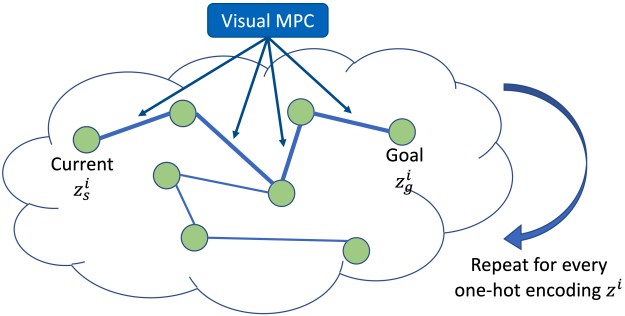

Figure 3: DORP Planning. Since the learned representation includes multiple one-hot encodings, we plan for each one-hot encoding one at a time. For each of the one-hot encodings $z^i$, we build a graph based on that subset of the representation. After finding the shortest path from current state to the goal state, we use MPC to reach the next planned state. See Section 3.2 for more explanations.

We propose an inductive bias in the similarity score function to encourage temporal consistency. Instead of a fully trainable matrix, we parameterize the similarity matrix $\psi$ to be diagonally dominant and diagonally positive (DDDP). This property biases the similarity score to be high when the key embedding $\tilde{z}_{t+k}$ is close to the query embedding $z_t$ and maximized when $\tilde{z}_{t+k} = z_t$. In other words, this incentivizes the representations of positive pairs to share as many one-hots as possible. We reparameterize the weight matrix as $e^{\psi_0}(-\sigma(\psi_1) + \alpha I_{l \times l})$ where $I_{l \times l}$ is an identity matrix, $\alpha$ is a positive constant, $\sigma$ is a sigmoid function, $\psi_0$ is a trainable scalar, and $\psi_1$ is a trainable $l$-by-$l$ matrix.

## 3.2 ABSTRACT PLANNING

A critical component in planning is a connectivity graph that decides whether two discrete representations are connected. One naive solution is to build such a graph from data – all observed representations as nodes and all observed transitions as edges. However as the number of independent entities increases, the amount of data required to generate the nodes and edges in order to cover the state space increases exponentially. To solve this, we propose to reduce the planning problem by exploiting factorization (see Figure 3).

1. First, embed the current image $o_s$ and goal $o_g$ as $z_s = q_\theta(o_s)$ and $z_g = q_\theta(o_g)$.

2. Pick a random one-hot index $i$ s.t. the current state's one-hot $z_s^i$ differs from the goal's one-hot $z_g^i$.

3. Build a graph based on how this one-hot transits in the data (ignoring all other one-hots).

4. Plan a path $z_1^i, ..., z_g^i$ from $z_s^i$ to $z_g^i$. If this does not exists, then we fail the task.

5. Execute the low-level controller following the next target $z_k^i$, $k = 1, \cdots, g$, while keeping other $z^j$ the same.

6. If it doesn't reach $z_k^i$, remove the edge from the graph and redo the planning in step 4.

7. If it succeeds, follow the next target until it reaches $z_g^i$.

8. Repeat step 2 until all the one-hots match.

9. Finally, execute the low-level controller to the goal image $o_g$.[1]

Our proposed method assumes that each latent one-hot can be manipulated independently to solve the task. This assumption can hold various settings. For example, imagine a task that requires reorganizing furniture in a spacious room, and each objects is represented as a separate one-hot. Here, we can manipulate each object separately, and since there is plenty of empty space in this scenario, we can easily plan around objects to avoid collisions.

This assumption does not always hold; along the same scenario, when the space is more cluttered, objects are more likely to collide, and the proposed fully-factorized planning may fail even though a solution exists (incomplete). **We extend this planning algorithm by building a graph on a random set of one-hot indices at a time in step 2**. When can choose this set to contain all the one-

---

[1] An implicit assumption of temporal consistency in the learned discrete representations is required.

hots the method is equivalent to the full graph search (complete but expensive). Thus, our extended planning method can trade between completeness and efficiency.

## 3.3 LOW-LEVEL CONTROL

The low-level controller is the key component in order to follow abstract plans and reach the goal. For this reason, the controller must be flexible in achieving multi-objectives. Model-predictive controller (MPC) achieves this by modeling the dynamics of the world from past data and minimizing with the total predicted cost of horizon $T$ at run-time: $a_t^*, ..., a_{t+T-1}^* = \arg\min_{a_t, ..., a_{t+T-1}} \mathbb{E}\left[\sum_{i=1}^{T} \gamma^t \hat{c}_{t+i}\right]$. The predicted cost is computed by applying a known cost function on the predicted outcome of a $T$-step action sequence from the current observation $o_t$.

MPC requires the world model and the costs. For the world model, we implement a stochastic video prediction model (Babaeizadeh et al., 2017). In our framework, we define two cost functions for reaching abstract goals and for reaching the final goal observation. The first cost function is defined to be 1 if the current embedding exactly matches the goal embedding (all the one-hots match) and 0 otherwise: $c_{\text{lat}}(o_t, z') = \mathbb{1}[q_\theta(o_t) == z']$. The second cost function is defined by its L2 loss in the observation space to the goal: $c_{\text{obs}}(o_t, o_g) = ||o_t - o_g||$. This cost is versatile for reaching nearby observations particularly to reach the goal observation once it has reached the goal code.

## 4 RELATED WORK

**Object Discovery** Recent work has studied unsupervised object segmentations from visual inputs (Burgess et al., 2019; Greff et al., 2019), and entity-factorized representations and models for predictions (Kipf et al., 2019). However, these studies have not been demonstrated to directly solve the task. In this work, we take a step further to evaluate the representations in downstream tasks. While other work (Veerapaneni et al., 2020; Watters et al., 2019) shows how the MPC agents benefit from object-factorized models, they require shaped rewards for the tasks. In contrast, we consider a long-horizon task in which the agent only receives its reward when it has reached the goal. A large body of work in computer vision has studied unsupervised video object segmentation (Aloimonos et al., 1988; Bajcsy, 1988; Bajcsy et al., 2018; Gibson, 2014; Fitzpatrick, 2003; Hausman et al., 2015; Kenney et al., 2009) and unsupervised object detection (Kwak et al., 2015; Wang et al., 2019; Xiao & Jae Lee, 2016). However, semantic segmentation and object detection might not be the correct representation for the end-task. In this work, we learn representations that are more ready to use for planning without explicit segmentation or detection.

**World models** Much previous work has applied generative models of the world to visual control tasks (Watter et al., 2015; Ha & Schmidhuber, 2018; Hafner et al., 2019). Other work (Ma et al., 2020; Okada & Taniguchi, 2020; Yan et al., 2020) leverages a contrastive objective to learn a latent world model. These methods however require dense reward signals for most tasks. Visual foresight methods (Finn et al., 2016; Ebert et al., 2018) demonstrate impressive results on real robots using unshaped reward such as pixel distance to the goal, but it still remains limited to short-horizon object pushing tasks. In our work, we borrow these methods for low-level controllers. Asai & Fukunaga (2017) learn discrete abstraction of the system such as an 8-piece puzzle, but do not consider temporal abstraction. Kansky et al. (2017) demonstrate that discrete object-factorized representations can be used to learn logic-based transitions. In combination with powerful planning, it improves generalization and data efficiency. However, it assumes supervised ground-truth labels for representation learning. In this work, we aim to learn this in an unsupervised manner.

**Hierarchial RL** Recent work has tried to approach long-horizon visual planning by breaking down tasks using skills (Lynch et al., 2020; Kipf et al., 2019). Our approach is orthogonal and may deploy such action abstractions as a low-level controller. Other methods propose to plan subgoals and attempt to follow them either in the latent space (Kurutach et al., 2018; Nasiriany et al., 2019; Nair & Finn, 2019) or in the visual space (Savinov et al., 2018; Liu et al., 2020; Eysenbach et al., 2019). However, such methods require labor-intensive engineering to tune the threshold on when to move on to pursue the next subgoal (Savinov et al., 2018; Liu et al., 2020) because the observations or the latent states can never exactly match. Imagine bringing a chair from one office to another. It would be time consuming to match all the positions and orientations of the chair along the way. Rather we should care about rough area the chair has to go through in order to reach the goal. In our work, by learning the discrete codes, our method do not require such threshold as it exactly knows when

it has reached the target discrete representation. Additionally, it avoids aiming to match a specific waypoint observation. Instead of planning to the goal, Pertsch et al. (2020) predict intermediate images iteratively to construct a subgoal tree. However in order to train such prediction model a long-sequential training data are required. Instead our work can be applied to short-horizon or long-horizon trajectories.

## 5    EXPERIMENTS

We perform experiments aimed at answering the following questions: (1) Are DORP representations temporally-consistent for high-level planning? (3) Can DORP representations factorize objects that translate or change in appearances over time? (3) How does DORP compare to the visual planning SOTA in solving unseen long-horizon goal-directed tasks?

### 5.1    ENVIRONMENTS

We evaluate DORP on two main environments – object-rearrangement and key-room. All the observations are provided as color images of size $16 \times 16 \times 3$.

$k$-**object-rearrangement** Multiple ($k$) 2-by-4 blocks of different colors can each be manipulated independently by an agent. The tasks require the agent to manipulate these objects from a start configuration to a goal configuration. In each step, it can manipulate each block by one unit in one of the 4 directions. The agent collects data by randomly interacting with one of the objects at each timestep in a purely exploratory manner, without any goal in mind. In this environment our trajectory has length one that is the configuration is randomly reset after every step.

**key-room** Two variations of key-room depend on the number of rooms – key-corridor (6 rooms and a corridor) and key-wall (2 rooms. A key and agent are represented by 1 pixel. A key is placed in a fixed location in a room. If the agent steps on the key a door will be removed allowing the agent to enter a locked room. Each step the agent can move in one of the four directions by 1 unit. The agent collect data by a long random walk – 1500 steps for key-corridor and 1000 steps for key-wall. After each reset the agent will be placed randomly in one of the unlocked rooms. This environment is inspired by MiniGrid (Chevalier-Boisvert et al., 2018).

### 5.2    BASELINES

We choose the state-of-the-arts visual foresight (VF) (Ebert et al., 2018) implemented using SV2P architecture (Babaeizadeh et al., 2017) as our baseline for two reasons. First, VF only requires self-supervised data collection and is applicable to unseen tasks – thus aligning with goal-directed visual planning problems. Second, Visual Foresight is a low-level controller of DORP. By sharing the same video model as baseline, we show a direct improvement of abstract planning for temporally-extended tasks. Two optimization variations of VF – VF with random shooting (VF-RS) and VF with cross-entropy method (VF-CEM) – both share the same video prediction model. Their objective is to minimize its L2 distance from the current image to the goal image. While DORP deploys VF-RS for its low-level control, VF-CEM comparision are provided for an improved baseline. VF-RS randomizes 1000 trajectories and take the full action sequence that achieves the minimum cost. VF-CEM randomizes 1000 samples per iteration. We pick the top 2% to refit a distribution over sequence and repeat for 3 iterations.

To understand DORP representation learning improvement, we evaluate DORP when replacing its DDDP similarity matrix by an arbitrary weight (Oord et al., 2018) or an identity matrix (He et al., 2020) which have been used extensively in unsupervised representation learning. To understand the benefits of factorized planning, we replace it with the full graph planning with a maximum limit on the number of steps allowed.

### 5.3    QUANTITATIVE RESULTS

**Temporal consistency** We investigate DORP representations by color-coding the discrete embeddings on the configuration map of the environments. For visualization simplicity we choose 1-

---

[3]With the exceptions for 5 objects in which we evaluate VF-RS on 10 tasks, VF-CEM on 11 tasks, and DORP on 23 tasks due to time and compute limits.

| Algorithms | 1 object | | 2 objects | | 3 objects | | 5 objects | |
|---|---|---|---|---|---|---|---|---|
| | SR | # steps | SR | # steps | SR | # steps | SR | # steps |
| DORP | **1.00** | $25_{\pm 15}$ | **1.00** | $49_{\pm 22}$ | **1.00** | $75_{\pm 28}$ | **0.87** | $1091_{\pm 1010}$ |
| – arbitary weight | 0.92 | $26_{\pm 19}$ | 0.58 | $52_{\pm 16}$ | 0.26 | $52_{\pm 16}$ | - | - |
| – identity weight | **1.00** | $34_{\pm 12}$ | 0.94 | $18_{\pm 7}$ | 0.48 | $83_{\pm 23}$ | - | - |
| – full graph | **1.00** | $25_{\pm 15}$ | **1.00** | $56_{\pm 17}$ | **1.00** | $76_{\pm 26}$ | 0.00 | max |
| VF-RS | 0.83 | $19_{\pm 19}$ | 0.58 | $53_{\pm 25}$ | 0.25 | $93_{\pm 24}$ | 0.00 | max |
| VF-CEM | 1.00 | $12_{\pm 15}$ | 0.60 | $47_{\pm 25}$ | 0.30 | $89_{\pm 27}$ | 0.10 | max |

Table 1: Comparison. We evaluate DORP in $k$-object-rearrangement tasks by measuring success rates across 50 sample tasks[3]. DORP is able to successfully solve most of the tasks as we increase the number of objects. We observe that other methods' performances degrade rapidly. We denote 'max' when the agent has reached the limit on the number of steps without reaching the goal in 5-object-rearrangment this limit is 1400.

object-rearrangement to visualize the learned code map. In this environment we have a single one-hot code for representing the object. We demonstrate that DORP learns temporally consistent representations in which two states from the same discrete code are connected by a short-horizon controller (see Figure 5(a)). Similarly, we observe the same property in key-wall environment (see Figure 6).

**Object factorization** We evaluate the behavior of the one-hot codes by changing one property of the agent at a time such as positions or whether the agent has a key. In Figure 4(c) we observe that by moving one object at a time only at most one one-hot code changes its value $k$-object-rearrangement. In key-room, we formulate the discrete representation as follows: of the two output one-hot latents, the first is set to a size $z_0$ and the other is set to size 2. Our aim is for one latent one-hot to encode the agent's abstract state and for the other binary latent to encode the key's abstract state. In Figure 6, we demonstrate that the learned abstract representations are factorized as desired by observing (1) the larger one-hot changes when the agent moves with no key interactions and (2) the binary one-hot changes when removing and adding the key while maintaining the agent position.

**Weight Ablation** We study the effect of the proposed similarity matrix against commonly-used similarity matrices (Oord et al., 2018; He et al., 2020) in unsupervised representation learning. In Figure 5, we demonstrate that DORP is able to exploit more available latent code and therefore helping the latent space to be more temporal consistent.

## 5.4 QUANTITATIVE RESULTS

**Long-horizon planning** We test the agent by its ability to solve unseen goal-directed tasks. In $k$-object-rearrangement, we can study the effect of the increase of the length and complexity of the tasks by increasing the number of objects. In Table 1, we demonstrate that as the number of objects increases DORP is able to succeed in most of the tasks. While other methods' performances degrade quickly. Note that when faced with 5 objects DORP employs the extended planning version by grouping the 5 one-hots into two groups of 2 and 3 one-hots. By considering more that one object at a time, the agent is able to perform non-myopic planning and achieve 87% success rate. We also evaluate DORP on two variations of key-room. In our evaluations, we find that DORP solves key-wall with a single wall and door successfully in 9 of 10 tasks. When using a ground-truth dynamics model in place of an SV2P model, DORP solves Key-Room successfully in 37 of 40 tasks, and Key-Corridor in 33 of 40 tasks.

## 6 CONCLUSION AND FUTURE DIRECTIONS

In this paper, we propose an unsupervised discrete representation learning method for long term planning, which can extract high level abstract states that are good for planning in an purely unsupervised manner. We demonstrate DORP's effectiveness over other methods on challenging long horizon tasks. We note that our method generate approximate plans more tractably and hence trades off optimality for efficiency, like other state abstraction works (Savinov et al., 2018; Liu et al., 2020; Eysenbach et al., 2019). Those approximate solutions can be used to initialize a model-free policy which can be later fine-tuned to reach optimality. We leave this as our future work.

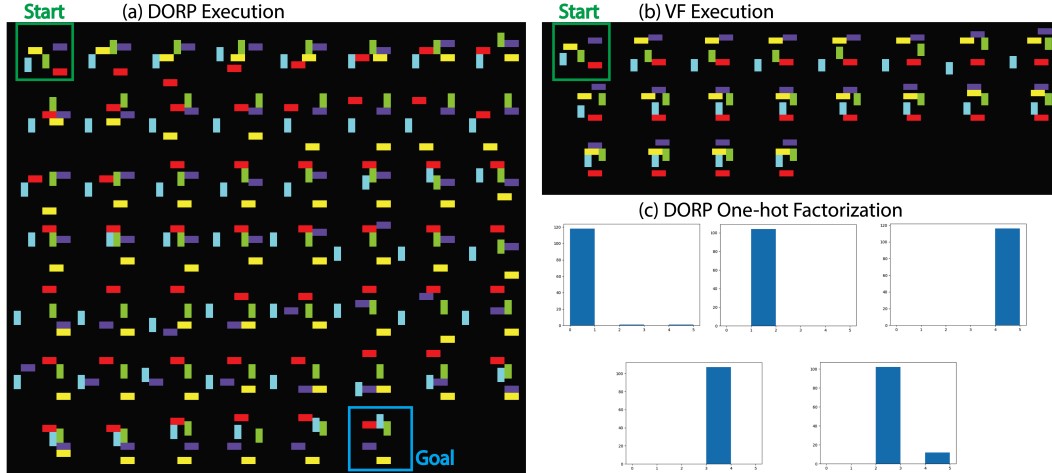

Figure 4: DORP in 5-object-rearrangement. In (a), we present DORP with random unseen start and goal images which require temporal-extended planning. In (b), when presenting the same task to VF-CEM we find that the objects are stuck in an awkward configuration where 4 blocks (except the purple) are blocking to reach their goal positions. Finally, in (c), we visualize the representation factorization by randomly moving one of the five object while maintaining the positions of the others. We plot a histogram of which one-hots have been changed per object. We find that with high probability only the one-hot that corresponds to the moving object is modified.

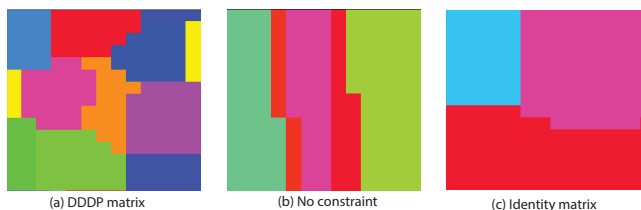

Figure 5: Discrete Embedding Comparison. We visualize the color codes of different object positions in 1-object-rearrangement per similarity matrix type. Each color shows different discrete code. The one-hot embedding has size 16 for all settings. In (a), we find that DORP discrete representation is temporally-consistent, i.e., two states that map to the same embedding are connected by a short sequence of actions. In (b), when using an arbitrary weight matrix the embedding is less temporally consistent. In (c), when using an identity matrix the, the embedding uses only 3 out of 16 available codes

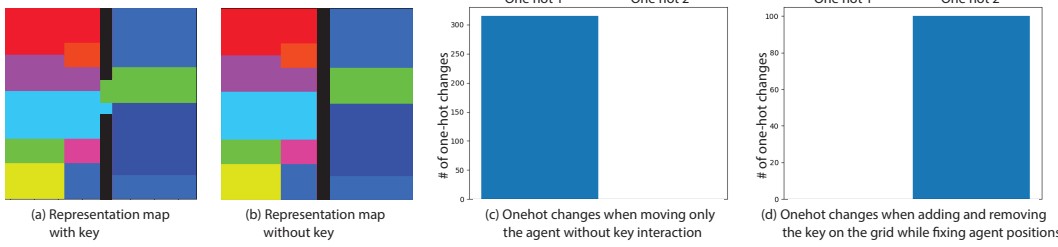

Figure 6: Key-Wall Representations. In (a) and (b), we demonstrate the discrete code of the agent at different positions. Each color represents the same code. The grids in black are invalid states (the wall that blocks the agent and separates the two rooms). We demonstrate temporal consistency in the latent space both when the object has the key as not. In (c) and (d), we confirm that two one-hot codes are factorized by observing that only one one-hot changes when removing the key while maintaining the agent position and only the other one-hot changes when the agent moves without interacting with the key.

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
