# OpenReview forum: "Discrete Predictive Representation for Long-horizon Planning"
_ICLR.cc/2021/Conference — Reject_

### Official Review · AnonReviewer3 · 2020-10-28

**Rating:** 4
**Confidence:** 4

**Review:**

This paper tackles the problem of long horizon visual planning, with the aim of of being able to plan actions to reach distant goals. This is a well studied problem, and like prior work this method considers the setting where the agent is given an offline dataset of interaction, which it learns from to be able to reach new goals (specified by a goal image). The method first does unsupervised representation learning, where it learns a discrete representation of images (using contrastive predictive coding (CPC) with a discrete latent variable using Gumbel Softmax). Using a set of discrete latent variable, it then builds a set of graphs which connects these discrete states based on the collected experience, and derives a planning procedure to find a path in the graph which reaches the goal.

Pros:
The problem of long horizon visual planning to goal images is important, and the paper does a good job of motivating it and describing the prior work in the area. The idea the paper presents is also intuitive and well motivated - while several recent works have explored building graphs and searching in these graphs to find subgoals, doing so over all experience can be costly, and discrete representations can aid in more compact graphs.

Furthermore, the proposed technique for learning this discrete representation, CPC but using a set of discrete latents trained with Gumbel Softmax is intuitive and to the best of my knowledge novel. And given a set of these discrete one hot encodings, the proposed inductive bias on the similarity function to encourage matching as many one-hots as possible seems to help performance.

Cons:
My main criticisms are on (1) the generality of the proposed planning procedure/representation learning, and (2) the thoroughness of the experimental comparisons.

(1):
First, while the proposed representation learning procedure seems to work quite well on the block world setting used in this paper, I have doubts about its ability to scale to more realistic environments. Since the proposed architecture learns a discrete latent variable for each channel feature map, it seems natural that this would easily learn to separate different multicolored blocks on a static background into different one hot encodings representing their position. But in more challenging domains like robotic manipulation or ego-centric visual navigation (which have been studied in the prior work), learning this discrete representation seems substantially more challenging, and its not clear that objects would be so cleanly separated.

Furthermore, these objects being cleanly factorized seems critical to the subsequent planning procedure. The idea of building a separate graph for each of the i one-hot encodings, and planning along a single one hot encoding at a time seems very specific to the domains used for evaluation. In general given a discrete representation of state, shouldn't the planning graph consider connections between this entire discrete state? The current planning algorithm (as the paper mentions) is dependent on the assumption that each latent one-hot can be manipulated independently. Not only does this in general not hold, it also implicitly assumes that the learned representation successfully factorizes the state space, which as mentioned above can be non trivial in harder environments.

(2):
The experimental evaluation is limited, in both domains and comparisons. First, there are many relevant works which the paper cites, but does not compare to. For the types of problems the paper looks at visual foresight is not the state of the art baseline. Methods like [1,2,3,4,5,6,7] (all of which the paper cites) would make for better points of comparison. Additionally, since a big part of the papers contribution is the method for learning the discrete representation, the paper should also compare to other techniques for learning such a discrete representation, such as [8,9].

Also as mentioned above, the experimental domains are visually simple, and designed in a way which is very particular to this method. I think showing results on more general problems like robotic manipulation or egocentric navigation are necessary to understanding if this approach really works.

Overall, this paper provides some interesting and well motivated ideas, but has some limitations in the generality of the proposed method and the thoroughness of the experimental evaluation.

[1] Kurutach et al. Learning plannable representations with causal infogan. 2018.
[2] Nasiriany et al. Planning with goal-conditioned policies. 2019.
[3] Nair et al. Hierarchical foresight: Self-supervised learning of long-horizon tasks via visual subgoal generation. 2019.
[4] Savinov et al. Semi-parametric topological memory for navigation. 2018.
[5] Liu et al. Hallucinative topological memory for zero-shot visual planning. 2020.
[6] Eysenbach et al. Search on the replay buffer: Bridging planning and reinforcement learning. 2019.
[7] Pertsch et al. Long-horizon visual planning with goal-conditioned hierarchical predictors. 2020.
[8] Caron et al. Deep Clustering for Unsupervised Learning of Visual Features. 2019.
[9] Van den Oord et al. Neural Discrete Representation Learning. 2018.

---

### Official Review · AnonReviewer1 · 2020-10-28
**Discrete Representations for Long-Horizon Planning -- Need better evaluations and baselines**

**Rating:** 4
**Confidence:** 4

**Review:**

This paper presents a method that combines learning discrete representations together with planning using graph search to solve long horizon tasks from vision. The approach works by generating data via random exploration and trains a representation encoder on this data. This network extracts objects from the observations and passes them through a CNN and shared encoder to generate one-hot encodings; these encodings are concatenated to generate the discrete representation. The encoder is trained via a contrastive learning objective with a similarity matrix that encourages nearby states to share similar encodings, thereby encouraging spatial and temporal abstractions. Next, these representations are combined together with an abstract planner to generate a sequence of waypoints to the goal. This is done by creating a graph of transitions from the collected exploratory data and search within this graph — this is executed for each encoding at a time with the assumption that the task can be solved by moving each object independently of the others. Finally, a low-level controller is used for reaching the waypoints and final goal, this is done via MPC on an action-conditional predictive model that generates future observations given the current state and goal. The approach is tested on two simple planar planning tasks where the agent has to solve k-object arrangement and open a room with a key respectively.

Pros:
1. This paper tackles an important problem of representation learning for long-horizon planning tasks. It proposes a standard bi-level architecture of a planner and low level controller but it nicely brings together different ideas from prior work such as the visual foresight based low level controller and graph search based high level planner.
2. A few novel points include the use of the Gumbel Softmax for learning discrete one-hot representations, the use of a DDDP matrix to encourage spatial and temporal similarity of learned encodings, and the factorised planning schema that takes advantage of the one-hot encodings to decompose the planning problem into tractable sub problems.
3. The presented results show that the approach scales well as the number of objects in the scene are increased. Even for rearrangement of 5 objects the method is able to get a performance close to 90% compared to baseline methods.

Cons:
While the proposed approach is well motivated there a few design choices that severely limit its applicability and the evaluations, baselines and results are a bit weak.
1. The approach is only tested on two visually simple 2D scenes where the task of learning a discrete object-centric representation is quite straightforward. This makes it hard to evaluate the strength and generalisation of the proposed approach — how well would the approach generalise to scenes that are more complex? It would be useful to have a discussion in the paper in this regard.
2. Looking at Fig 2., the encoder uses an “Extractor” that seems to generate bounding boxes and produce features of these boxes. What is this extractor and how is it trained? No details are provided regarding this network. If this is a network that is already trained to crop objects in the scene then the job for the encoder is quite straightforward — it just has to learn an object classifier which should be trivial to do in the current setting where the scenes are quite simple visually. If this is the case then it is quite obvious that the network would learn object-centric representations.
3. The presented baselines use the low-level controller to reach the final goal with a cost that is the error in image space. This is a very weak baseline (especially due to the cost function) that is bound to get stuck in local optima for the presented problems. An alternative baseline could be a method that uses a latent space error term similar to the proposed approach but with a dense latent that is not a one-hot vector. Such a latent could also be trained via a contrastive loss and has a better chance at succeeding on the proposed task — thereby forming a stronger baseline. It can also better elucidate the strength of a discrete representation for abstract planning.
4. The success of the proposed approach is highly dependent on the availability of good exploration data. It is not clear how this data is collected currently (a random walk is mentioned but more details would be useful). Also, how much data is collected? Is there an analysis on how well the approach does with varying amounts of data use? Similarly, how are the different networks trained? It would be great if more details are provided.
5. As mentioned in the paper, the method assumes that the planning problem can be factored into sub-problems where the task is to just move individual objects in the scene. While an alternative formulation of randomly sampling “k” one-hots and planning in this graph is provided this is still a fundamental assumption that limits the applicability of the proposed approach.

Other comments:
1. In Fig 5.a) there seems to be some wraparound between the different discrete codes. Why does this occur?
2. The planning algorithms use a very large number of samples/trajectories for these tasks. Is this needed?

Overall, the approach combines several interesting ideas but needs better baselines and evaluations. More details also need to be provided regarding the training, network architectures etc. I would suggest a weak reject.

---

### Official Review · AnonReviewer4 · 2020-10-28
**Interesting idea for task planning needs more than preliminary experiments**

**Rating:** 4
**Confidence:** 4

**Review:**

The paper presents learning discrete encoding (as one-hot vectors) given a scene image to represent various semantics about the scene with an aim to perform graph search over these representations in performing high level tasks.

The research direction is quite relevant and the motivation make a good case for it. Related work is adequately discussed. This is a promising line of work but in its current state a bit premature due to two main related reasons: a mismatch between the motivation/goal discussed early on and the validation performed; and the experiments are conducted on very small and simplistic game like setups.

- The two main environments studied are game like and too simplistic. While these can help in building intuition without more sufficiently complex experiments it is hard to understand the boundaries within which this approach can be used. Input images are 16x16 which is quite small. Several new environments like AI2THOR are around to study task planning, would the proposed method be able to tackle any such scenarios? More task planning baselines (particularly non-learning based) should be considered. What if a goal image is not available as the target but the task is provided by other means?

- Building the initial representation by pure exploration seems prohibitively expensive for complex setting, would any efficient alternative work?

- Learning 'low level control' is referenced several time in the paper but the experiments have no mention of it or are designed to explore/study this.

Other comments:

- It wasn't too clear in the initial read until much later in the paper, what 'object-factorized' really meant

- Problem statement in 2.1 is quite vague and grounding it in an example would aid in understanding.

- The underlying approach is a bit hard to fully understand. An explanation of 3.2 along with a simple example in parallel would help (maybe walkthrough a full iteration of the approach).

---

### Official Review · AnonReviewer2 · 2020-11-04
**Interesting and promising approach, but lacks motivation, particularly through experiments.**

**Rating:** 4
**Confidence:** 4

**Review:**

This work presents Discrete Object-factorized Representation Planning (DORP), which learns a discrete representation from videos with an enforced temporal consistency. This representation can then be planned over through a sequence of small alterations to the discrete embedding, which are then executed via MPC. DORB is demonstrated to solve long-horizon tasks and learn representations that consider objects and their properties.

The paper is clear, though there are several typos on the paper, including the two words of the intro (“In the future”). The paper should be proofread carefully. The work is interesting, particularly the efficient search and graphical representation of a state space to plan high-dimensional and long-horizon problems. Such approaches are quite promising and the approach is well founded. The figures and demonstrations qualitatively demonstrate the learned representation’s properties nicely.

The planning process proceeds by altering the goal state one-hot’s a single index at a time until the initial state is reached. A few aspects and consequences of this planning are not clear:
- How does one know if a path does not exist at step 4?
- How often are one-hot’s not represented in the data? Can density of the latent discrete embeddings be encouraged in any way or does this naturally arise?
- “We extend this planning algorithm by building a graph on a random set of one-hot indices at a time in step 2.” This should be written in the algorithm description for clarity.

My main issue is the experiments, in terms of (1) motivating the approach, (2) baseline comparison, and (3) difficulty.
1) The primary contribution is the framework for learning this discrete embedding and planning over it, but the need for a discrete embedding is never fully considered. This should be ablated over with a continuous latent space, as has been used previously e.g, in https://arxiv.org/abs/1807.10366, Watter 2015, Hafner 2019 and https://arxiv.org/pdf/1912.01603.pdf and related works.
2) The results should compared to methods such as Eysenbach 2019 and Savinov 2018 as these use similar representations, which are discrete via their use of actual datapoints.
2) The examples are nice to specifically show properties of the learned space and visualize them, but the problems are not particularly realistic. How would DORP work with a more continuous task like a robot pick and place with multiple objects for example?

Minor notes:
- Why is the 1 object result for VF-CEM not bolded in Table 1? It seems most efficient in steps and has a 1.00 SR.
- A video would be helpful of both learning the space and planning.

---

### Decision · Program_Chairs · 2021-01-07
**Final Decision**

**Decision:**

Reject

**Comment:**

This paper explores a foundational problem in AI around learning abstractions that allow for easier planning.  The work proposes a specific procedure for learning temporally abstract, discrete representations in which it becomes tractable to perform graph-based search.  Evaluation is performed on two 2D tasks where a goal is specified visually and the system must produce the actions to achieve this target state/observation.

The reviewers were in a uncommonly tight consensus as to their evaluation of the paper (all 4).  All reviewers essentially expressed that the motivation of the paper was solid but that the domains considered were too simplistic for validation of this class of approach, especially in light of substantial previous work in the area that was not adequately captured in the baseline comparisons.  The authors did not respond to the reviewers.

My decision is to reject the paper.